# Quenching protein dynamics interferes with HIV capsid maturation

Mingzhang Wang[1,2], Caitlin M. Quinn[1,2], Juan R. Perilla [1,3], Huilan Zhang[1,2], Randall Shirra Jr.[2,4], Guangjin Hou [1,2], In-Ja Byeon[2,4], Christopher L. Suiter[1,2], Sherimay Ablan[5], Emiko Urano[5], Theodore J. Nitz[6], Christopher Aiken[2,7], Eric O. Freed[5], Peijun Zhang[2,4,8], Klaus Schulten[3], Angela M. Gronenborn[2,4] & Tatyana Polenova [1,2]

Maturation of HIV-1 particles encompasses a complex morphological transformation of Gag via an orchestrated series of proteolytic cleavage events. A longstanding question concerns the structure of the C-terminal region of CA and the peptide SP1 (CA–SP1), which represents an intermediate during maturation of the HIV-1 virus. By integrating NMR, cryo-EM, and molecular dynamics simulations, we show that in CA–SP1 tubes assembled in vitro, which represent the features of an intermediate assembly state during maturation, the SP1 peptide exists in a dynamic helix–coil equilibrium, and that the addition of the maturation inhibitors Bevirimat and DFH-055 causes stabilization of a helical form of SP1. Moreover, the maturation-arresting SP1 mutation T8I also induces helical structure in SP1 and further global dynamical and conformational changes in CA. Overall, our results show that dynamics of CA and SP1 are critical for orderly HIV-1 maturation and that small molecules can inhibit maturation by perturbing molecular motions.

[1] Department of Chemistry and Biochemistry, University of Delaware, Newark, DE 19716, USA. [2] Pittsburgh Center for HIV Protein Interactions, University of Pittsburgh School of Medicine, 1051 Biomedical Science Tower 3, 3501 Fifth Ave., Pittsburgh, PA 15261, USA. [3] University of Illinois, Theoretical and Computational Biophysics Group, Urbana, IL 61801, USA. [4] Department of Structural Biology, University of Pittsburgh School of Medicine, 3501 Fifth Ave., Pittsburgh, PA 15261, USA. [5] HIV Dynamics and Replication Program, Center for Cancer Research, National Cancer Institute, Frederick, MD 21702, USA. [6] DFH Pharma, Gaithersburg, MD 20886, USA. [7] Present address: Department of Pathology, Microbiology and Immunology, Vanderbilt University Medical Center, Nashville, TN 37232, USA. [8] Present address: Division of Structural Biology, University of Oxford, The Henry Wellcome Building for Genomic Medicine, Headington, Oxford, OX3 7BN, UK. Mingzhang Wang, Caitlin M. Quinn, and Juan R. Perilla contributed equally to this work. Correspondence and requests for materials should be addressed to J.R.P. (email: jperilla@udel.edu) or to A.M.G. (email: amg100@pitt.edu) or to T.P. (email: tpolenov@udel.edu)

Human immunodeficiency virus (HIV), the causative agent of acquired immune deficiency syndrome, has claimed more than 35 million lives so far. An essential step in the HIV-1 lifecycle, necessary for generating infective virions, occurs upon virus release from an infected host cell[1] and involves processing of Pr55[Gag] (Gag) into its constituent domains by the viral protease (PR). This sequential cleavage cascade is termed maturation. Gag is one of the principal polyproteins of HIV-1 and its constituent domains are (from N-terminus to C-terminus): matrix (MA), capsid (CA), spacer peptide 1 (SP1),

**Fig. 1 a** Schematic diagram of the HIV-1 Gag sequential cleavage and virus maturation process. RNA was omitted for clarity. **b** CA–SP1 cleavage. The ribbon diagram of the CA monomer is shown with the CypA loop and MHR highlighted in orange and the SP1 region depicted as a dotted blue line. The T8I mutation in SP1 mimics the presence of maturation inhibitors (MI) in abolishing SP1 cleavage. **c** A cryo-EM image of CA–SP1 tubular assemblies. Scale bar, 50 nm. **d–h** Cryo-EM reconstruction of CA–SP1 assemblies. **d** Surface rendering of the of CA–SP1 3D density map, low-pass filtered to 8 Å resolution. The density map (contoured at 2σ) is colored in orange and blue for CA–CTD and CA–NTD, respectively, viewed along (top) and perpendicular to (bottom) the tube axis. **e** MDFF fitting of three CA hexamers (PDB code 4XFX, gold, magenta, and blue ribbons) into the density map. **f** Superposition of the ribbon diagrams of three CA molecules at the trimer interface (green, PDB code 3j34) onto the equivalent model for the CA–SP1 trimer interface (gold, magenta, and blue). **g, h** Comparison of the dimer (**g**) and trimer (**h**) interfaces in CA assemblies (green) to those in CA–SP1 assemblies (gold, magenta, blue). **i** The variabilities among the six CA molecules in CA (top) and CA–SP1 (bottom) assemblies. **j** Assembly assay of CA–SP1(T8I) NL4-3 and CA NL4-3 for different concentrations of NaCl. **k** TEM images of tubular assemblies of CA(A92E) and CA(A92E)–SP1 variants

**Table 1 CSI 2.0 secondary structure prediction based on experimental chemical shifts for CA–SP1(T8I) NL4-3, CA (A92E)–SP1 NL4-3, and CA–SP1 HXB2**

| Subdomain | Residue | CA–SP1 (T8I) NL4-3 | CA(A92E)–SP1 NL4-3 | CA–SP1 HXB2 |
|---|---|---|---|---|
| CA | V221 | C | C | C |
| | G222 | C | C | C |
| | G223 | C | C | C |
| | P224 | C | -- | -- |
| | G225 | C | -- | -- |
| | H226 | H | C | C |
| | K227 | H | C | C |
| | A228 | H | C | -- |
| | R229 | H | C | C |
| | V230 | H | H | -- |
| | L231 | H | H | C |
| SP1 | A1 | H | H | H |
| | E2 | H | H | H |
| | A3 | H | C | H |
| | M4 | H | C | -- |
| | S5 | H | C | -- |
| | Q6 | H | C | -- |
| | V7 | C | -- | -- |
| | I8/T8 | C | C | C |
| | N9 | C | C | -- |
| | P10 | C | -- | -- |
| | A11 | C | C | C |
| | T12 | C | C | -- |
| | I13 | C | C | -- |
| | M14 | C | C | -- |

Designations C-coil and H-helix are as described in the Methods section. TALOS-N[75] secondary structure predictions and predictions for additional CA and CA–SP1 constructs are provided in the Supplementary Tables 2, 3. "--" indicates predictions that are less reliable due to missing chemical shifts

with BVM resistance[27, 28], suggesting that this region may be the BVM binding site. For PF96, resistance mutations reside at the CA–SP1 boundary, the MHR, as well as CA residue 201[29]. Furthermore, propagation of replication-defective, PF96-dependent MHR, and SP1 mutants generated the T8I mutation in SP1 as a second-site compensatory mutation[29, 30], and the T8I variant was found to affect CA–SP1 processing (Fig. 1b) as well as possess reduced infectivity (~15% of WT). T8I corrects assembly defects of MI-dependent MHR mutations[29, 31], seemingly phenocopying the effect of MI binding by stabilizing the immature lattice[31].

Despite extensive studies, atomic level structural data in the presence of MIs have not been reported. Here, we used an integrated approach, combining MAS NMR, solution NMR, cryo-EM, and molecular dynamics (MD) simulations to investigate how the MI BVM interferes with CA maturation, and how the T8I mutation in the SP1 region influences the structure and dynamics of the CA–SP1 maturation intermediate. We present evidence that BVM and related analogs bind at and perturb the conformation of the CA–SP1 junction. We demonstrate by MAS NMR that the SP1 region is stabilized in a helical conformation in CA–SP1(T8I) tubular assemblies and that structural changes are induced throughout the entire CA domain. Our cryo-EM density map reveals a relaxed CTD trimer interface and establishes that the SP1 region is largely missing, indicating that it likely lacks a stable helical conformation. This finding agrees with prior results from MAS NMR[16], which showed that SP1 exists in a dynamic helix–coil equilibrium. In the CA–SP1(T8I) mutant, the motions in SP1 are greatly reduced and the helical content is increased. Functionally, this escape mutation renders the CA non-infections in the presence of Cyclophilin A, which is normally required for virus infectivity, but does not interfere with ordered tubular assembly. The experimental isotropic solid-state NMR chemical shifts of tubular CA–SP1(T8I) assemblies are in excellent agreement with those calculated from MD trajectories of a CTD–SP1 (T8I) hexamer.

Taken together, our results suggest that CA maturation occurs through modulation of protein dynamics, and that small-molecule MIs function by interfering with these motions.

## Results

**In the cryo-EM structure of CA–SP1, SP1 is not a helix.** Purified CA(A92E)–SP1 protein assembles well into helical tubes under conditions identical to those used previously for CA(A92E) (Fig. 1c), adopting a wide range of helical symmetries. The CA (A92E) mutant was used since less bundling of tubes is present, resulting in higher resolution cryo-EM data than for the wild-type (WT) CA[4], in addition to attenuating the CypA-binding loop dynamics[32]. Among these, the tubes of the most abundant helical symmetry (−8, 14) were selected for three-dimensional (3D) reconstruction, using the iterative real-space helical reconstruction method (see Zhang et al.[33] for information regarding helical symmetry). The cryo-EM structure of the CA(A92E)–SP1 tubular assembly was determined from 33 tubes at 8 Å resolution (Fig. 1d–i), with a gold standard fourier shell correlation (FSC) of 0.143 (Supplementary Fig. 1). All α-helices in the resulting density map were well-resolved, which permitted accurate MDFF fitting of three neighboring CA hexamers (Fig. 1e, f) and analysis of the interactions at the dimer and trimer interfaces[4].

In both CA(A92E) and CA(A92E)–SP1 tubular assemblies, the atomic models of the CA–NTDs superimpose well (Cα RMSD = 1.2 Å), and the CA–CTD dimer interfaces are also similar in both structures (Cα RMSD = 1.6 Å). As previously noted for the CA (A92E) assemblies, different H9 crossing angles are present along specific helical arrays in the dimer interfaces, although the extent of variability is less pronounced in the CA(A92E)–SP1 assemblies

nucleocapsid (NC), spacer peptide 2 (SP2), and p6. In the mature virion, the CA forms the genome protecting protein shell, which contains ~216 CA hexamers and 12 pentamers in a fullerene type arrangement[2–4]. The CA protein monomer is divided into an N-terminal domain (NTD) and a C-terminal domain (CTD), connected by a flexible linker[5, 6]. A highly conserved sequence in the CA protein, the major homology region (MHR), plays a critical role in assembly, maturation, and infectivity[7, 8].

The final step in the maturation cleavage cascade is the removal of the SP1 peptide from CA[9, 10], which triggers rearrangement of an immature lattice into the final mature conical shape[11, 12]. The details of this conformational rearrangement are still elusive. Three pathways have been proposed (Fig. 1a): (i) gradual reorganization of the immature lattice to form the mature CA (displacive)[13], (ii) de novo reassembly from a pool of CA monomers[14, 15], with SP1 acting as a molecular switch and inducing disassembly of the immature lattice[11, 16, 17], and (iii) a sequential combination of displacive and de novo processes[18].

In the context of the immature CA, cryo-EM studies have suggested the presence of a six-helix bundle for the SP1 region[15, 19, 20], and the isolated CTD–SP1 protein can also form a six-helix bundle under certain crystallization conditions[21]. In tubular assemblies of the CA–SP1 maturation intermediate, magic angle spinning (MAS) NMR reveals the SP1 region as a dynamic random coil[16]. Maturation inhibitors (MIs), such as Bevirimat (BVM), PF-46396 (PF96), and their analogs[22–25] are thought to bind to the CA–SP1 junction, thereby blocking cleavage and stabilizing the immature lattice[22, 26]. This suggests MIs as a promising class of anti-retroviral therapeutics. Sequence polymorphism within the first 6–8 amino acids of SP1 is associated

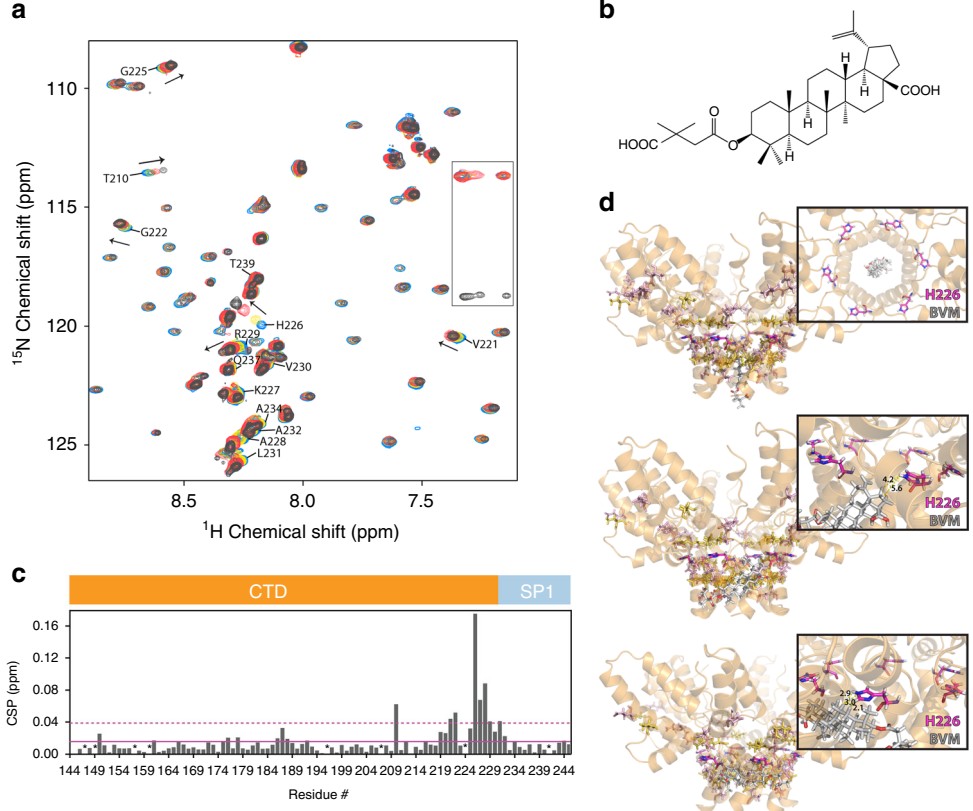

**Fig. 2** Solution NMR and rigid body docking of Bevirimat binding to CTD–SP1. **a** Superposition of selected region of the $^1H$-$^{15}N$ HSQC spectra of 0.4 mM CTD–SP1 in the absence (blue) and presence of 1.11 mM (yellow), 2.22 mM (red), and 3.33 mM (black) Bevirimat at 298 K. Representative CTD–SP1 resonances are labeled with residue names and numbers. Folded arginine side chain Nε resonances are enclosed in the rectangle. **b** $^1H$,$^{15}N$-combined chemical shift perturbations (CSP) for CTD–SP1 resonances in the absence and presence of Bevirimat (3.33 mM), respectively. The solid and dashed horizontal lines indicate the average chemical shift change (0.016 ppm) and the sum (0.038 ppm) of the average change plus one S.D., respectively. Proline residues and the unassigned S149 positions are marked with *. Note the large CSPs observed for resonances of residues 219–233 are indicative of the interaction between Bevirimat and CTD–SP1. **c** Chemical structure of Bevirimat. **d** Binding poses of BVM (gray) in the CTD–SP1 hexamer (X-ray structure PDB code 5I4T) resulting from rigid body docking. The insets are expansions, illustrating the relative positioning of BVM and the H226 residue (colored magenta). Residues with CSPs > 0.04 and >0.02 ppm are colored yellow and light pink, respectively. Residues in the CTD–SP1 junction whose resonances experience CSPs > 0.02 ppm are located in the binding interface identified in the docking studies

($C\alpha$ RMSD = 0.9 Å between dimers) than in the CA(A92E) assemblies ($C\alpha$ RMSD = 2.7 Å). In contrast, the trimer interface in the CA(A92E)–SP1 assembly (Fig. 1f, h, between gold, magenta, and blue monomers) is less compact than in the CA(A92E) assembly (Fig. 1f–h, green) similar to the trimer interface observed in the CA–SP1–NC structure[34]. In addition, the linker between the NTD and CTD is conformationally less variable in CA(A92E)–SP1 than in CA(A92E) ($C\alpha$ RMSD = 0.8 and 2.1 Å for CA monomers in a CA(A92E)–SP1 hexamer and a CA(A92E) hexamer, respectively) (Fig. 1i). This suggests that SP1 limits the conformational space available for positioning the CTD. Interestingly, some PF96-resistance mutations were mapped to this region[29, 31].

For the SP1 peptide itself, no detectable electron density was present, indicating that SP1 is likely conformationally or dynamically disordered and does not form a stable, unique helical structure in the tubes. This result is consistent with our prior findings from MAS NMR[16].

The secondary structure of a truncated SP1 peptide was previously investigated in solution by NMR and was reported as α-helical in organic solvents as well as at high concentrations in aqueous solution[17, 35]. Sequence-based protein structure prediction also suggested the presence of an α-helix at the CTD–SP1 junction[36]. In a solution NMR investigation of a longer CTD–SP1–NC construct, SP1 exhibited a slight helical propensity[37], in agreement with our current solution NMR studies (see below). For tubular assemblies of WT CA–SP1 and mutant CA(A92E)–SP1, the MAS chemical shifts suggest a predominantly random coil conformation while for tubular CA–SP1(T8I) assemblies, the MAS shifts are indicative of a significant helical propensity for residues 226–231 (CTD) and 1–6 (SP1), see Table 1 and Supplementary Table 1.

**BVM binds at the CTD–SP1 junction in solution.** For the isolated WT CTD–SP1 protein in solution, $^1H$-$^{15}N$ HSQC spectra obtained in the absence and presence of BVM (Fig. 2a) or DFH-055 methanesulfonate, a "second-generation" BVM analog with significantly improved antiviral activity[25] (Supplementary Fig. 2) permitted delineation of the amino acids affected by MI binding, based on NMR chemical shift changes which, as is well appreciated, are exquisitely sensitive to changes in local environment, conformation, and/or dynamics[38]. In the absence of the MI, the solution chemical shifts of the amide resonances of the CTD tail (residues 223–231) and the SP1 subdomain (residues 1–14) region exhibit little dispersion with narrow linewidths, indicating that this region is largely disordered. A careful inspection of the $C\alpha$ and $C\beta$ chemical shifts, however, reveals that small, positive secondary chemical shifts can be discerned, suggesting a slight helical propensity (Supplementary Fig. 2). Upon MI binding,

resonances of the CTD tail residues as well as those of SP1 residues are perturbed, with the largest changes observed for H226 (0.18 ppm), K227 (0.07 ppm), and A228 (0.09 ppm) (Fig. 2b). The large shift for the H226 amide resonance is of interest and suggests that BVM binding changes the conformation at position 226. Indeed, this notion is consistent with our

docking results, as discussed below. We also note that V221 resides in a β turn between helix 11 and the random tail, and BVM may stabilize an interaction between helix 11 and preceding residues. In addition, a substantive change was also noted for the T210 resonance (0.06 ppm). This chemical shift change could be caused by an allosteric structural change that is the result of BVM

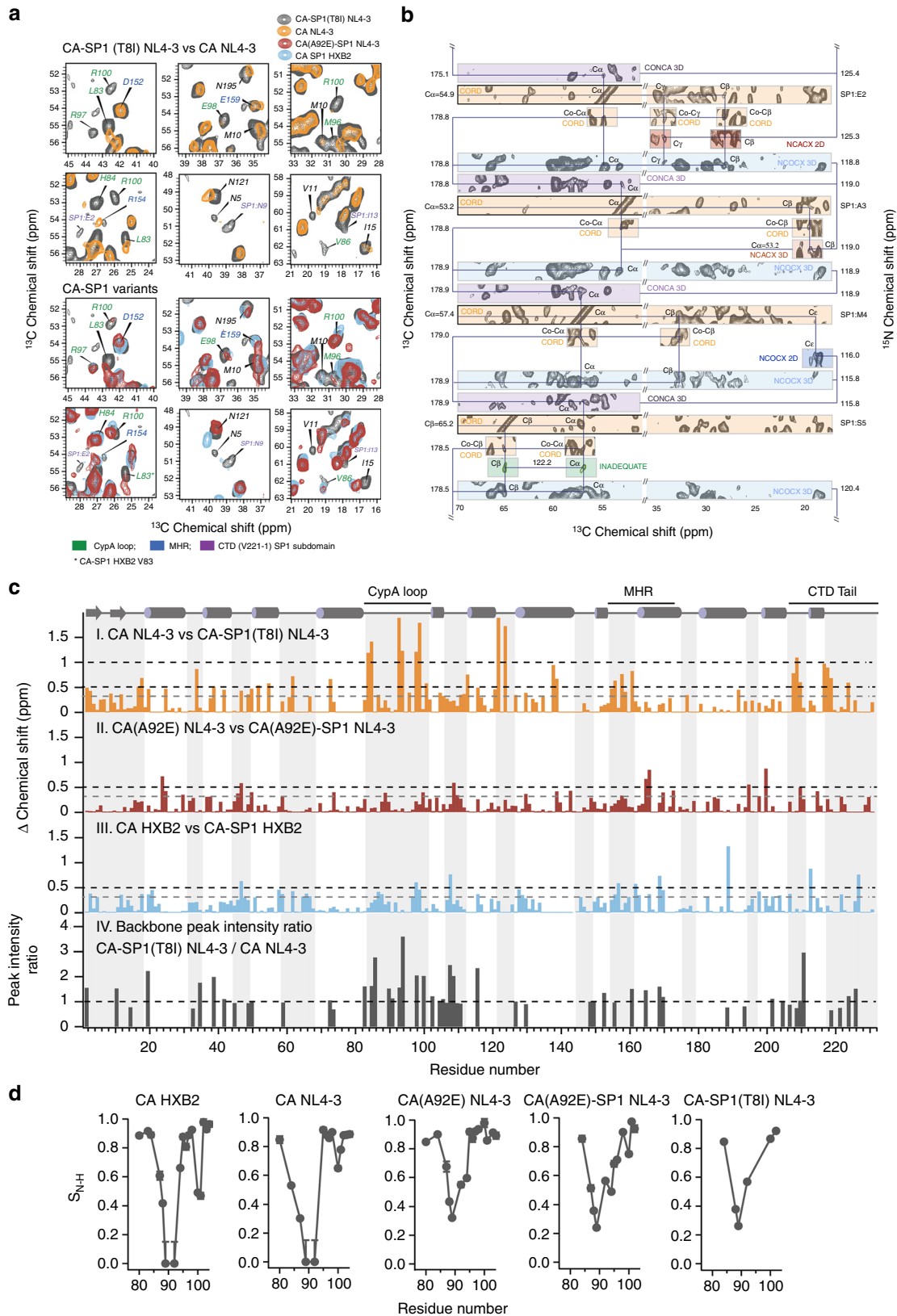

blocking a transient interaction between helix 8 and SP1, as suggested by the MD results (see below). These findings suggest that both BVM and DFH-055 interact with the CTD–SP1 junction in the isolated CTD–SP1 construct.

To further examine the BVM binding mode, we performed rigid docking of BVM (Fig. 2c) to the recently reported X-ray structure of CTD–SP1[21] (PDB code: 5I4T). We note that this structure captures a rare six-helix bundle conformation at the CTD–SP1 junction (which includes CA residues 221–231 and SP1 residues 1–7), while SP1 residues 8–14 appear to be disordered[21]. The docking results indicate that BVM indeed binds at the CTD–SP1 junction, consistent with the NMR data. These docking simulations suggest three possible BVM binding modes (Fig. 2d): (i) binding inside the pore formed by the six-helix SP1 bundle; (ii) binding to a single CTD–SP1 chain; (iii) binding simultaneously to two CTD–SP1 chains. All three binding modes are possible based on the cryo-EM results on immature CAs[15, 39]. The current solution NMR results indicate that in solution the most likely mode is BVM binding simultaneously to two CTD–SP1 chains. As shown in Fig. 2d, in this binding mode H226 is in close proximity to BVM, in accord with the observed chemical shift perturbation (CSP) of the amide resonance. Gratifyingly, the residues in the CTD–SP1 junction whose resonances experience CSPs of >0.02 ppm are also those comprising the binding interface according to the docking calculations.

Extending our rigid body docking calculations to the recently published CA–SP1 model derived by CryoEM[39] suggests the presence of the same binding modes as the ones obtained for docking of BVM to the X-ray structure of CTD–SP1 [21] (Supplementary Fig. 6).

Additional experiments would be required to determine the binding modes of BVM to the assembled CA–SP1. Unfortunately, these experiments have proven challenging to date due to the low solubility of BVM and its analogs in aqueous solutions.

**The T8I mutation induces a helical conformation in SP1.** CA and CA–SP1 proteins form tubular assemblies (Fig. 1c, k), which recapitulate the CA's hexameric lattice[4, 6], and these tubes yield remarkably high-quality MAS NMR spectra[16, 32]. From a suite of 2D and 3D homo- and heteronuclear correlation spectra, complete site-specific resonance assignments for the CA–SP1(T8I) tubes were obtained, including the CTD tail and the SP1 subdomain (Fig. 3a, b). Both the CTD tail and the SP1 region, containing the T8I mutation, give rise to strong, well-defined peaks in dipolar-based correlation spectra, revealing that these segments of the protein are rigid on the microsecond timescale. In contrast, in our previous work on CA–SP1 HXB2 and CA (A92E)–SP1 NL4-3, resonances of residues in the CTD tail and SP1 were absent in dipolar-based experiments, indicating that in these WT and WT-like proteins the equivalent regions exhibit

substantial motions on a timescale of tens of microseconds[16]. The experimental chemical shifts indicate that the T8I mutation in SP1 results in increased helical content (Table 1), with residues H226(CA)–Q6(SP1) forming a contiguous helix. In contrast, in CA–SP1 HXB2 and CA(A92E)–SP1 NL4-3 the majority of the residues are in a random coil conformation.

**The T8I mutation induces allosteric changes in the CA domain.** Remarkably, the T8I mutation in the SP1 subdomain also influences the conformational and dynamic properties in other regions of the CA domain (Fig. 3), in addition to affecting residues in the CTD tail and SP1 subdomain close to the mutation site. As we have shown previously[16], the presence of SP1 modulates the structure of the CA domain, and in the tubular assemblies of CA–SP1 from the HXB2 strain and of CA (A92E)–SP1 from the NL4-3 strain, a number of residues exhibit sizeable CSPs (>0.3 ppm), compared to the corresponding CA assemblies (Fig. 3c, panels II and III). Even more strikingly, in the CA–SP1(T8I) mutant, significantly larger chemical shift changes (>1–1.5 ppm) are introduced for many residues, indicating dramatic conformational differences due to this maturation-inhibiting mutation (Fig. 3c, panel I). These conformational differences are linked to a change in the overall dynamics of the CA domain, and the microsecond–millisecond motions are reduced, as evidenced by increased peak intensities throughout the CA–SP1(T8I) spectrum (Fig. 3c, panel IV). Regions where major conformational and dynamics effects are observed include the N-terminal β-hairpin, the Cyclophilin A-binding loop (CypA loop, loop 4/5), and loop 6/7 in the NTD, in addition to the MHR and loops 9/10 and 10/11 in the CTD. The overall reduction in dynamics throughout the T8I mutant protein is also supported by the smaller number of resonances in scalar-based correlation MAS NMR spectra of tubular CA–SP1(T8I) assemblies, compared to those of CA(A92E)–SP1. In these experiments, only residues that undergo dynamics on the micro- to millisecond timescales result in observable signals (Supplementary Figs. 3, 4). Thus, overall the T8I change reduced the dynamics of CA, rendering it conformationally less flexible.

**Helix–coil equilibrium of SP1 is modulated by T8I mutation.** Our prior and current MAS NMR results indicate that the SP1 subdomain is dynamic and in a predominantly random coil conformation in CA–SP1 HXB2 and CA(A92E)–SP1 NL4-3 assemblies[16], while it adopts a stabilized helix in assemblies of CA–SP1(T8I) from the NL4-3 strain. To further assess the structure and motions in the SP1 region, we performed simulated tempering MD calculations over 15–24 µs of a hexameric subunit of CTD–SP1 HXB2 as well as the CTD–SP1(T8I) HXB2. Remarkably, and in complete agreement with the MAS NMR data, these simulations revealed a highly mobile SP1 region for

**Fig. 3** MAS NMR data for CA–SP1 assemblies. **a** Expansions of the superposition of 2D[13]C-[13]C correlation spectra for CA and CA–SP1 variants highlighting SP1 resonances and residues exhibiting chemical shift or peak intensity changes. Chemical shift perturbations are present for resonances of several residues in the CTD–SP1 tail (V221-to end), CypA loop, MHR, and NTD β-hairpin. Top: CA NL4-3 (orange) and CA–SP1(T8I) NL4-3 (gray). Bottom: CA (A92E)–SP1 NL4-3 (red), CA–SP1 HXB2 (blue), and CA–SP1(T8I) NL4-3 (gray). **b** Sequential assignment walk for a stretch of CA–SP1(T8I) residues (SP1 residues E2–S5). Note the large intensity of the associated resonances, indicative of reduced dynamics. Their chemical shifts are consistent with increased helicity, compared to WT CA–SP1. **c** Chemical shift changes between CA and CA–SP1 plotted along the linear amino-acid sequence. I, II, III: sum of [13]Cα,[15]N chemical shift perturbations (CSP): CA NL4-3 and CA–SP1(T8I) NL4-3 (orange), CA(A92E) NL4-3 and CA(A92E)–SP1 NL4-3 (brown), CA HXB2, and CA–SP1 HXB2 (blue). Significant CSPs of >0.5 ppm are observed for the CypA loop, the MHR region, and the CTD tail. CSP values < 0.3 ppm (dashed gray) are negligible and within experimental and systematic error. IV: Plot of the [13]Cα-[15]N backbone peak intensity ratio (CA–SP1(T8I) NL4-3/CA NL4-3; gray) vs. residue number (for non-overlapping peaks with resonance assignments in the 2D NCA MAS NMR spectra). Residues with a peak intensity ratio >1 possess attenuated motions on micro- to millisecond timescales. **d** Sequence dependence of CypA loop dynamics for five different HIV-1 CA and CA–SP1 variants. [1]H-[15]N dipolar order parameters are plotted vs. residue number for CypA loop residues in CA HXB2, CA NL4-3, CA(A92E) NL4-3, CA (A92E)–SP1 NL4-3, and CA–SP1(T8I) NL4-3. The CA–SP1(T8I) mutant exhibits the same order of magnitude attenuation of loop dynamics as previously observed in the CA(A92E) escape mutant[32]

the WT hexamer, exhibiting a dynamic equilibrium between random coil and helical conformations, with the helix representing a minor, transient form (Fig. 4a–c). In addition, frequent, short lived contacts between SP1 and helix 8 of the MHR were observed.

In order to gain a quantitative understanding of the experimental isotropic MAS frequencies of CA–SP1 assemblies, chemical shifts were calculated with the data-based program SHIFTX2[40] for every MD frame and each of the six WT CTD–SP1 chains. Averaging the shifts for the MD trajectory over

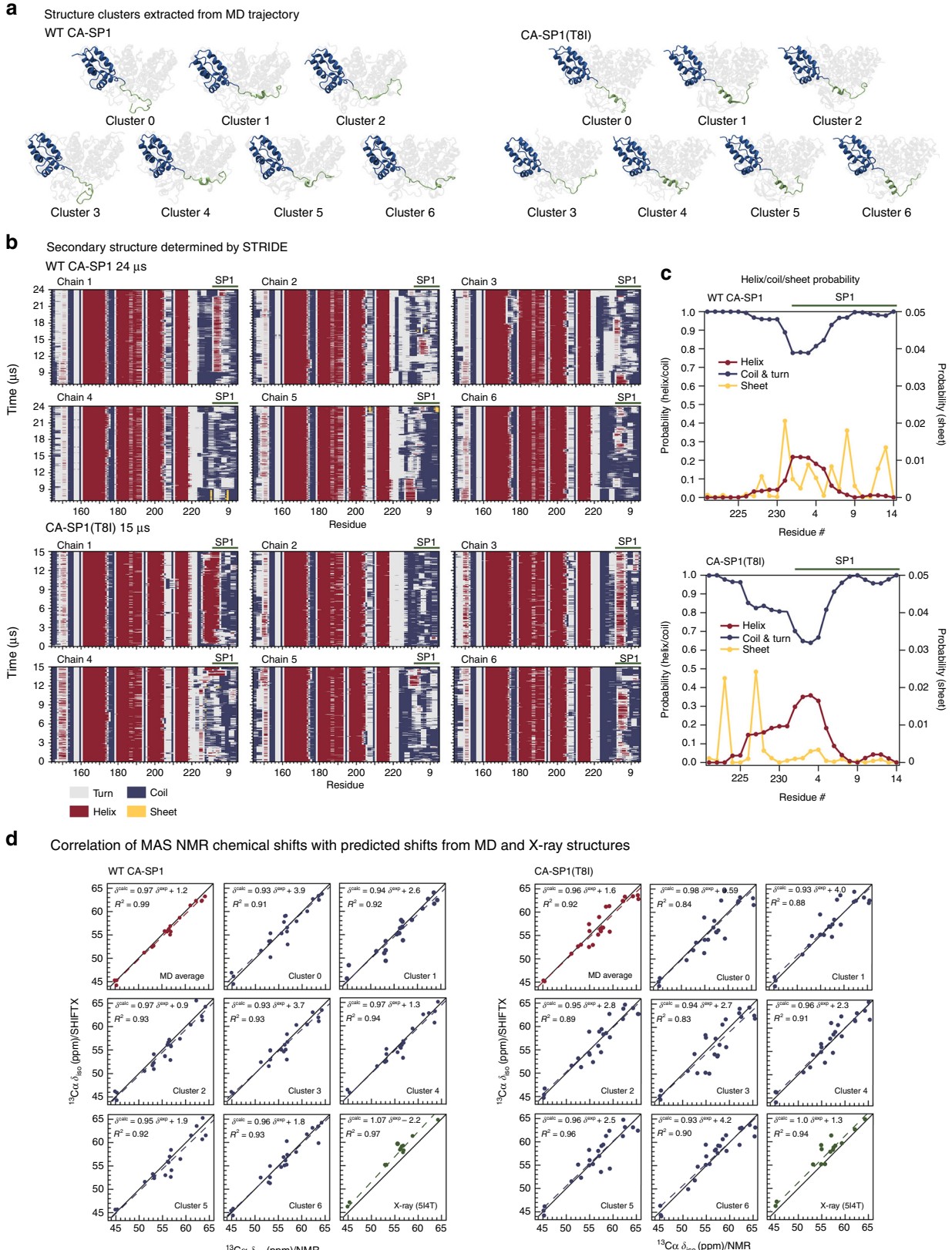

all frames and all chains of the hexamer yielded excellent agreement between calculated and experimental shifts (Fig. 4b). Interestingly, substantial scatter and poor correlations are observed when shifts are calculated for any individual sub-population (cluster) of the MD trajectory (Fig. 4a), determined using the partition around medoids protocol[41]. This result unequivocally not only corroborates our prior experimental MAS NMR-based conclusion that the CTD tail and SP1 subdomain exhibit dynamic helix–coil conformations, but also provides a quantitative understanding of the contributions of the individual cluster types to the experimental chemical shifts. Recent dynamic nuclear polarization experiments, which result in small, but narrow lines for a minor helical SP1 sub-population lent further support to this notion[42]. Moreover, the SP1 six-helix bundle in the recent X-ray structure of the CTD–SP1[21] may be stabilized by the presence of a helix-inducing solvent (Bis-Tris propane) during crystallization, and possibly by the presence of an N-terminal His tag, which may force SP1 into the minor helical state, fortuitously captured in the crystal. In this regard, it is interesting to note that chemical shifts calculated using the X-ray diffraction data from the CTD–SP1 crystals (PDB: 5I4T) agree well with the MAS chemical shifts for the CA–SP1(T8I) mutant assembly, but less well with those for WT CA–SP1 assemblies (Fig. 4d), supporting our assertion that a highly mobile SP1 region is present in the WT CA–SP1 assemblies.

MD simulations of the CA–SP1(T8I) mutant suggested reduced dynamics and increased helical propensity. Secondary structure propensity plots generated using STRIDE from the MD trajectory show that SP1 forms stable helices that persist throughout the entire length of the MD trajectory in two of the six chains (Fig. 4b), and the SHIFTX2-predicted chemical shift values are in excellent agreement with experimental values (Fig. 4d). Interestingly, the strongest correlations are observed for structures with greater helical content, including the average over all frames of the MD trajectory, as well as clusters 4 and 5 individually. This corroborates our MAS NMR-based observation that the predominant major structure is helical in the T8I mutant and suggests that longer MD simulations may be needed to possibly observe the six-helix bundle.

## Discussion

A significant novel finding from the work presented here is the direct demonstration of the key role that dynamics plays in regulating CA maturation. The dynamic helix–coil equilibrium in the CTD–SP1 subdomain of CA–SP1 emerged as the pivotal feature in the mechanism of the final maturation step. We posit that the predominantly random coil conformation of the CTD–SP1 region in WT CA–SP1 assemblies, present in this dynamic equilibrium, is essential for correct and efficient processing of the Gag maturation intermediate, permitting the formation of mature conical CAs in infectious virions.

Another noteworthy result is the atomic level understanding of the effects caused by BVM binding and the T8I mutation in SP1

on the structure and dynamics of CA–SP1. Our solution NMR data in combination with docking provide clear experimental evidence that BVM and its more potent DFH-055 analog bind directly to the CTD–SP1 junction, as postulated previously[23, 43]. The residues at the CTD–SP1 junction identified by NMR to be interacting with BVM upon binding are also those that comprise the BVM binding interface according to the docking calculations.

The most pronounced effect of the T8I mutation which mimics BVM binding, is greatly attenuated dynamics and increased helicity of residues H226(CA)–Q6(SP1), which form a contiguous helix in the mutant. Indeed, binding of BVM to this region will shift the helix–coil equilibrium toward the helical conformer.

Another surprising effect of the T8I mutation is the degree of allosteric modulation imparted on the conformation and dynamics of CA, affecting both domains, NTD and CTD. The most dramatic changes are associated with motions in the CypA loop. This loop is highly flexible in the assembled CA, and modulation of dynamics in this loop plays a functional role in HIV-1's escape from CypA dependence[32]. Remarkably, the MAS NMR experiments unequivocally show that nano- to micro-second timescale dynamics of the CypA loop are dramatically attenuated in assemblies of the CA–SP1(T8I) mutant, as evidenced by significantly increased $^1$H-$^{15}$N dipolar order parameters compared to those in WT CA assemblies of the same NL4-3 strain (Fig. 3d). Interestingly, a similar, same order of magnitude reduction of loop dynamics was previously observed in the A92E and G94D escape mutants of CA[32]. Viruses bearing these mutations exhibit a dramatic loss in infectivity in vivo, which can be reversed by cyclosporin. In contrast, CA–SP1 assemblies without the T8I mutation exhibited increased dynamics in the CypA loop, compared to the corresponding CA assemblies[16], possibly indicating that NTD–NTD contacts may be altered by this SP1 mutation[15, 44].

Similarly, the N-terminal β-hairpin is stabilized in the CA–SP1 (T8I) assemblies, compared to WT CA assemblies. The conformation of the N-terminal residues in the CA–SP1 lattice is currently under debate: it has been hypothesized that the β-hairpin is present only in the mature CA[44], while other reports suggest that it is formed upon cleavage between the MA and CA domains[45, 46]. The conformation of the N-terminal β-hairpin influences the central intra-hexamer pore size and possibly CA's permeability to nucleotides[47]. Indeed, a chemical shift index (CSI) analysis of the MAS NMR chemical shifts indicates that the N-terminal residues form a β-hairpin in all CA–SP1 assemblies examined here (Supplementary Table 3), implying that CA–SP1 cleavage is not required for the formation of the β-hairpin or that the β-hairpin forms early during maturation. The T8I mutation also induces significant chemical shift changes for residues in loop 6/7, mutations of which affect HIV-1 replication. This loop resides at the trimer interface in the immature lattice[15], and the T8I-induced chemical shift changes may suggest the presence of altered trimer interface contacts. The observations of dramatic chemical shift changes and altered dynamics in the NTD for CA–SP1(T8I) are rather surprising, since CA–SP1 HXB2 and CA

**Fig. 4** MD simulations of the CTD–SP1 hexamer. **a** Clustering analysis of the MD trajectory of CA–SP1 WT (left) and CA–SP1(T8I) mutant (right) identified six major sub-populations. One chain is colored in blue (CA-CTD) and green (SP1) to illustrate the differences in secondary structure present in the SP1 region: predominantly random coil in CA–SP1 WT and significantly increased helical content in CA–SP1(T8I). **b** Stride plots of secondary structure for each chain along with the MD trajectories for CA CTD–SP1 WT (top), and CA CTD–SP1(T8I) mutant (bottom). **c** Helix/coil probability of the CTD tail (V221-end)SP1 subdomain averaged over the MD trajectories: CTD–SP1 WT (top) and CTD–SP1(T8I) (bottom). The expanded scale (0–0.05; right hand side) is shown for the "Sheet" content. **b**, **c** The CTD tail (V221-end)SP1 region exhibits a dynamic equilibrium between random coil and helical conformations in CA CTD–SP1 WT, whereas in the CA CTD–SP1(T8I) mutant the dynamics are greatly attenuated and the helical content is increased. **d** Correlation of MAS NMR chemical shifts and SHIFTX2-predicted shifts from the MD trajectory seeded from the X-ray structure (PDB: 5I4T): CTD–SP1 WT (left) and CTD–SP1 (T8I) (right). Note the remarkable agreement between the experimental and predicted shifts, indicating that the MD simulations accurately capture the conformational equilibrium in the assembled CA–SP1

(A92E)–SP1 exhibit much smaller chemical shift changes in the NTD compared to CA HXB2 and CA(A92E), respectively. Thus, the T8I mutation has a larger influence on the NTD conformation than the presence of the SP1 peptide alone. Taken together, the MAS NMR results clearly show that in assembled CA–SP1 (T8I), the conformation of the NTD is different compared to WT CA–SP1 or WT CA assemblies.

Chemical shift changes due to the presence of the T8I mutation are also observed throughout the CTD. They are associated with several residues in functionally relevant regions, such as the MHR (residues 153–172), and loops 9/10 and 10/11. Specifically, the T8I mutation introduces increased conformational and dynamic heterogeneity (multiple conformers are detected) into the highly conserved MHR region (Supplementary Figs. 3–5). The MHR is thought to be involved in stabilizing intra-hexameric interactions in the immature lattice[15]. Therefore, the T8I mutation and, by virtue of its phenocopying effect, BVM and related MIs may influence regions that undergo critical conformational rearrangements during maturation. Furthermore, replication defects resulting from PF96-dependent resistance mutations in the MHR can be compensated for by mutations in the CTD tail or SP1 subdomain (such as T8I), suggesting that an allosteric "cross-talk" between the SP1 peptide and MHR[29] may exist. Interactions between helices 9/10 and 10/11 have been suggested to play a role in the formation of the immature lattice, and the dimer interface (helix 9) in the immature lattice is different from that of the mature lattice[15, 20]. While we did not observe significant changes for nano- to micro-second timescale dynamics of residues in the loops connecting helices 9/10 and 10/11, chemical shift changes unequivocally indicate conformational differences, which may indeed be due to altered positioning of the adjacent helices. Overall, the perturbations we observe in the CA–SP1(T8I) assemblies suggest that MI binding affects the structure of those CTD regions that undergo critical conformational rearrangements during maturation.

The CA–SP1 tubes are a valuable in vitro model system to study SP1 dynamics, and we hypothesize that they represent an intermediate assembly state during maturation. Importantly, the motions we detected and quantified are an inherent feature of SP1 encoded by the primary sequence of the peptide rather than the morphology of the assembly, as is evident from the comparison of WT and the T8I mutant of SP1, and similar to the motions found in other subdomains of CA.[4, 16, 32, 48] Based on the combined evidences, we submit that the results on SP1 tubes used in our study reflect the behavior of SP1 in authentic viruses.

Taken together, our results suggest that interfering with the dynamic helix–coil equilibrium in the CTD–SP1 subdomains of CA–SP1 by small molecules or mutations in the SP1 subdomain may constitute a promising venue for the development of novel anti-HIV therapeutic strategies.

To conclude, assessing the conformational effects of BVM binding and dynamics changes in the maturation-inhibiting CA–SP1(T8I) mutant led to important mechanistic insights into SP1 cleavage inhibition by MIs. Significant structural differences are observed at the CTD–SP1 junction upon BVM binding, with affected residues becoming restricted in motions and exhibiting increased helical propensity in the maturation-inhibiting CA–SP1 (T8I) mutant, compared to WT CA–SP1 assemblies. The more rigid helical conformation adopted by the mutant may be a key impediment to proteolytic cleavage. Allosteric changes noted in NMR observables of the CA domain suggest that structural changes in the lattice of CA–SP1(T8I), which mimics the MI-bound form, play important roles. Our data reveal that the conformation and dynamics of diverse regions in CA–SP1(T8I) differ from the WT CA lattice, both in the NTD, such as the CypA loop and loop 6/7, as well as the CTD, for instance the MHR.

In contrast to CA–SP1(T8I), the SP1 subdomain of WT CA–SP1 exists in a highly dynamic helix–coil equilibrium, as shown by MAS NMR, cryo-EM, and MD studies. Such a flexible, dynamic conformation facilitates easy access of the PR to the cleavage site, permitting maturation to proceed. Since the WT CA–SP1 lattice resembles that of mature CAs, the maturation most likely proceeds via a displacive pathway or a sequential combination of displacive and de novo building processes.

Taken together, our integrated experimental–computational approach yielded an atomistic understanding of the underlying mechanistic details of HIV CA maturation and effects that MIs play in this important step. Future studies using similar approaches can now be applied to elucidate the molecular basis of action by "second-generation" BVM analogs[25, 49].

## Methods

**MAS NMR sample preparation and characterization.** HIV-1 CA and CA–SP1 proteins were expressed and purified essentially as reported previously[16]. The following sequence variants/mutants were studied: WT CA NL4-3, CA(A92E) NL4-3, WT CA HXB2, CA(A92E)–SP1 NL4-3, WT CA–SP1 HXB2, and CA–SP1 (T8I) NL4-3; where (A92E) and (T8I) designate mutations in the CA and SP1 regions, respectively; NL4-3 and HXB2 refer to strain variants. The CA(A92E) mutation was used since in previous work we showed that the A to E change prevents the tubes from aggregating, resulting in higher resolution cryo-EM data than for the WT CA[4], and attenuates the CypA loop dynamics[32]. Functionally, this mutation renders the CA non-infections in the presence of cyclophilin A, normally required for virus infectivity. The gene was amplified and ligated into pET21 vector using NdeI and XhoI sites. Escherichia coli Rosetta2 (DE3) cells were transformed with the different vectors. Uniformly $^{13}$C,$^{15}$N-enriched CA, and CA–SP1 proteins were expressed in modified M9 media, containing $^{15}$NH$_4$Cl and U-$^{13}$C$_6$-glucose as sole nitrogen and carbon sources, respectively. Expression was induced with 0.8 mM IPTG at 18–23 °C for 16 h. Cells were harvested by centrifugation, resuspended in 25 mM sodium phosphate buffer (pH 7.0) and opened by sonication. The lysate was centrifuged at 15,000 rpm at 4 °C for 1 h, the pH of the supernatant was adjusted to 5.8 with acetic acid, and the conductivity reduced by dilution to below 2.5 ms/cm. After a second centrifugation step at 15,000 rpm at 4 °C for 1 h, the final supernatant was loaded onto a cation exchange column and protein was eluted with a 0–1 M NaCl gradient in a buffer containing 25 mM sodium phosphate (pH 5.8), 1 mM DTT, and 0.02% NaN$_3$. Concentrated protein fractions were further purified over a size-exclusion column, equilibrated in 25 mM sodium phosphate buffer (pH 7.0), 1 mM DTT, and 0.02% NaN$_3$.

WT CA NL4-3 and CA–SP1(T8I) NL4-3 tubes were pre-assembled from 26 mg/ml protein solutions in 25 mM phosphate buffer (pH 5.5), containing 2.4 M NaCl, followed by incubation at 37 °C for 1 h and 4 °C overnight. Tubular assemblies of CA and CA(A92E)–SP1 NL4-3 and WT CA–SP1 HXB2 were prepared from 32 mg/ml protein solutions in 25 mM phosphate buffer (pH 5.5), containing 1 M NaCl[16]. The various CA and CA–SP1 assemblies were pelleted at 10,000 rpm and packed into Bruker 3.2 mm rotors.

The morphologies of CA and CA–SP1 assemblies were characterized by transmission electron microscopy (TEM). TEM analysis was performed with a Zeiss Libra 120 transmission electron microscope operating at 120 kV. Assemblies were stained with uranyl acetate (0.5–1% w/v), deposited onto 400 mesh, formvar/ carbon-coated glow discharged copper grids, and dried for 45 min in the air.

**Binding studies of maturation inhibitors to CTD–SP1.** BVM was purchased from Adooq Bioscience LLC (Irvine, CA) and DFH-055 methanesulfonate was a generous gift of DFH Pharma, Inc. We also investigated DFH-005 methanesulfonate, in addition to BVM, since it is more soluble than BVM and a "second-generation" BVM analog, which displays significantly improved antiviral activity relative to the parent compound[25]. Inhibitors were dissolved in methanol to prepare a 3.3 mM stock solution. Small aliquots (60 μl, 200 nmol) of the stock solution were transferred into 3 mm NMR tubes and completely dried by flushing with nitrogen gas. $^{13}$C,$^{15}$N-labeled CTD–SP1 (containing NL4-3 CA–CTD(144–231) and SP1) was prepared using the same procedure as for $^{13}$C,$^{15}$N-labeled CA–CTD[6]. All spectra were acquired at 14.1 T (600 MHz), except for the spectrum in the presence of 3.33 mM BVM, which was acquired at 18.8 T (800 MHz). For binding studies, a $^1$H-$^{15}$N HSQC reference spectrum was recorded using 0.4 or 1.1 mM $^{13}$C,$^{15}$N-labeled CA CTD-SP1 protein in 25 mM sodium phosphate buffer (pH 6.5), 1 mM TCEP, 0.02% sodium azide, and 7% D$_2$O (180 μl in a 3 mm NMR tube). Subsequent $^1$H-$^{15}$N HSQC spectra were acquired after transferring the reference sample to a 3 mm NMR tube containing an aliquot (200 nmol) of the dried inhibitor. This procedure was repeated one or two more times, resulting in total added inhibitor of 400 and 600 nmol, respectively. Since the inhibitors are poorly soluble in aqueous solutions, some visible precipitate was observed in the NMR tubes, precluding accurate determination of $K_D$ values. $^1$H,$^{15}$N-combined chemical shift changes were calculated using the equation $(\Delta\delta_{HN}^2 + (0.15 \times \Delta\delta_N)^2)^{1/2}$, with $\Delta\delta_{HN}$ and $\Delta\delta_N$ representing $^1$H and $^{15}$N CSP.

**MAS NMR spectroscopy.** MAS NMR experiments were performed on Bruker 20.0 and 14.1 T narrow bore Avance III spectrometers outfitted with 3.2 mm E-Free HCN probes. Some experiments for CA HXB2 and CA–SP1 HXB2 assemblies were performed at the National High Magnetic Field Laboratory (NHMFL) on a 21.1 T ultrawide bore magnet equipped with a Bruker Avance III spectrometer, outfitted with a 3.2 mm HXY sensitivity-enhanced Low-E MAS probe. MAS NMR spectra were collected at a MAS frequency of $14.000 \pm 0.005$ kHz, regulated by a Bruker MAS controller, and sample temperature of $4 \pm 0.1$ °C throughout the experiments using a Bruker temperature controller. $^{13}$C and $^{15}$N chemical shifts were referenced with respect to the external standards adamantane and ammonium chloride, respectively.

MAS NMR spectra of tubular assemblies of CA NL4-3, CA–SP1(T8I) NL4-3, CA(A92E) NL4-3, and CA(A92E)–SP1 NL4-3 were acquired at 20.0 T, operating at Larmor frequencies of 850.4 MHz ($^{1}$H), 213.8 MHz ($^{13}$C), and 86.2 MHz ($^{15}$N). The MAS frequency was 14 kHz. The typical 90° pulse lengths were 2.4–3.1 μs ($^{1}$H), 3.2–3.9 μs ($^{13}$C), and 3.5–4.0 μs ($^{15}$N). The $^{1}$H-$^{13}$C and $^{1}$H-$^{15}$N CP employed a linear amplitude ramp of 90–110% on $^{1}$H, and the center of the ramp matched to Hartmann–Hahn conditions at the first spinning sideband, with contact times of 1.1–1.5 and 1.5–1.8 ms, respectively. For 2D$^{13}$C–$^{13}$C CORD[50] spectra, the CORD mixing time was 50 ms, and $^{1}$H field strength during CORD was 12.5–13.5 kHz. For 2D NCA, 2D NCACX, and 2D NCOCX, band-selective magnetization transfer from $^{15}$N to $^{13}$C, the contact time was 4.3–5.5 ms. SPINAL-64 decoupling[51] (80–96 kHz) was used during the evolution and acquisition periods. For 3D RN symmetry-based DIPSHIFT and PARS experiments[52, 53], R12$_1^4$ symmetry sequences with an RF field strength of 84 kHz were applied for reintroducing $^{1}$H-$^{15}$N dipolar couplings. Some experiments of CA–SP1(T8I) NL4-3 assemblies were performed at 14.1 T, with Larmor frequencies of 600.8 MHz ($^{1}$H), 150.8 MHz ($^{13}$C), and 60.8 MHz ($^{15}$N). The typical 90° pulse lengths were 2.8 μs ($^{1}$H), 3.6 μs ($^{13}$C), and 4.5 μs ($^{15}$N). The $^{1}$H-$^{13}$C and $^{1}$H-$^{15}$N CP employed a linear amplitude ramp of 90–110% on $^{1}$H, and the center of the ramp matched to Hartmann–Hahn conditions at the first spinning sideband, with contact time of 0.8 and 1.4 ms, respectively. The CORD mixing time was 50 ms, and $^{1}$H field strength during CORD was 14 kHz. Band-selective magnetization transfer from $^{15}$N to$^{13}$C contact time was 5.0–6.5 ms. SPINAL-64 decoupling (89 kHz) was used during the evolution and acquisition periods.

All spectra were processed in TopSpin and with NMRpipe[54] and analyzed using SPARKY and CCPNMR[55]. For 2D and 3D data sets, 30°, 45°, 60°, or 90° shifted sine bell apodization followed by a Lorentzian-to-Gaussian transformation was applied in both dimensions. Forward linear prediction to twice the number of the original data points was used in the indirect dimension in some data sets followed by zero filling to twice the total number of points. Spectra of CA NL4-3 tubular assemblies were assigned by comparison with CA resonance assignments reported previously[16]. Spectra of CA–SP1(T8I) NL4-3 tubular assemblies were assigned by a sequential walk, using a combination of 2D CORD/direct-CORD, 2D/3D NCACX, 2D/3D NCOCX, 3D CONCA, and 2D J-INADEQUATE experiments.

**Simulations and calculations of NMR dipolar parameters.** SIMPSON (versions 1.1.2 and 2)[56] was used to perform the numerical simulations of $^{1}$H-$^{15}$N dipolar lineshapes. For RN-DIPSHIFT experiments, 986 pairs of [α, β] angles were generated with the ZCW algorithm, and five γ angles that resulted in a total of 4930 angle triplets were used to calculate a powder average, as reported previously[32]. About 320 pairs of [α, β] angles were generated with the REPULSION algorithm, and 16 γ angles that resulted in a total of 5120 angle triplets were applied to calculate a powder average for RN-PARS experiments. The experimental and processing parameters (i.e., MAS frequency, Larmor frequency, RF field strength, number of T1 points, zero filling, line broadening, finite pulse lengths, etc.) were utilized for extracting the best-fit dipolar parameters with simulations. For all the 3D RN-DIPSHIFT/PARS spectra, a series of home-written C++ programs and shell scripts were used to process automatic extractions of dipolar lineshapes for the residues in CypA loop (H84-S102), iteratively with manual inspections, to ensure the correctness of the assignments and extractions.

**Molecular dynamics simulations.** All-atom MD simulations were performed in explicit solvent using the TIP3P water model[57] and the CHARMM22 STAR force field[58]. The simulations were carried out both on general-purpose supercomputers using NAMD 2.11[59] and on the special-purpose supercomputer Anton[60]. All simulations using NAMD were carried out with periodic boundary conditions and constant particle number, constant temperature, and constant volume.

Simulations on Anton were carried out using the simulated tempering method[61–63] and employed an integration time step of 2.0 fs, with short-range forces evaluated at every time step, and long-range electrostatics evaluated at every second time step. Short-range non-bonded interactions were cut off at 17 Å; long-range electrostatics were calculated using the k-Gaussian Split Ewald method[64] with a 32 × 32 × 32 grid. The temperature rung and associated weights were calculated using the Park method[65], and are presented in Supplementary Tables 4 and 5 for WT and the T8I mutant, respectively. Expectation values of experimental observables were weighted using the simulated tempering weighted histogram analysis method (ST-WHAM)[66].

**Calculations of chemical shifts from MD trajectories.** For WT CA CTD–SP1, 7394 frames were extracted from 7–24 μs MD trajectory. For CA CTD–SP1(T8I) mutant, 5860 frames were extracted from 0–15 μs MD trajectory. SHIFTX2[40] was used to calculate $^{13}$Cα, $^{13}$Cβ, $^{13}$C′, and $^{15}$N chemical shifts for each frame of each single chain. An averaged chemical shift for each atom over all six chains from all the frames of the MD trajectory was calculated and compared with the corresponding experimental shift. The STRIDE program was used to predict the secondary structures for every frame from atomic coordinates[67]. We classified the different secondary structure outputs into four types: helix, β-sheet, coil, and turn. The helix classification included an α-helix, a 3$_{10}$ helix, and a π Helix. The β-sheet classification included an extended configuration and an isolated β-sheet hydrogen bond formation.

**Cryo-EM specimen preparation and data collection.** Full-length CA(A92E)–SP1 NL4-3 proteins (3.95 mg/ml) were diluted to 2.2 mg/ml in high salt buffer (1 M NaCl, 50 mM Tris pH 8.0) and incubated at 37 °C for 1 h for tubular assembly. 2.5 μl of sample was applied to the carbon side of a glow discharged perforated Quantifoil grid (Quantifoil Micro Tools, Jena, Germany), followed by application of 3 μl of low salt buffer (100 mM NaCl, 50 mM Tris pH 8.0) on the back side of the grid, and blotting, from the back side, with a filter paper, before plunge-freezing in liquid ethane, using a manual gravity plunger. Low-dose (~23 e$^{-}$/Å$^{2}$) projection micrographs of tubes embedded in a thin layer of vitreous ice were collected on a FEI Titan Krios microscope (FEI Corp., Hillsboro, OR.) operated at 300 kV with parallel illumination. Images were recorded on Kodak SO163 films at a nominal magnification of 59,000, using under-focus values ranging from 1.0 to 3.5 μm. The best micrographs were selected and digitized using a Nikon super coolscan 9000 ED scanner (Nikon, Japan) at an optical resolution of 4000 dpi. The pixel size on specimen is 1.1 Å/pixel.

**Image processing and 3D reconstruction.** More than 200 films of tube images, possessing a wide range of diameters and helical symmetries, were collected. About 33 tubes with (−8, 14) symmetry were selected and boxed into segments of size 512 × 512 pixels with a shift of two subunits along the helical axis by the EMAN BOXER program[68]. The defocus value of each micrograph was evaluated using the program CTFFIND3[69]. An initial density map was reconstructed from binned segments of 256 × 256 pixels using iterative helical real space reconstruction[33]. The refined helical parameters corresponded to a rise of 13.46 Å and a rotation angle of 128.88°. The preliminary map was further refined using Frealign[70] without binning. During the refinement, helical symmetry and full-CTF correction were applied. A total of 14,000 helical segments were included in the final reconstruction. The FSC curve was calculated using gold standard procedure from two independent half data sets, indicating a resolution of 8.0 Å at FSC value of 0.143.

**Molecular dynamics flexible fitting.** A helical tube of mature-like CA(A92E)–SP1 NL4-3 was constructed by docking the MDFF-derived hexameric structure (PDB: 3J34), using Chimera, into the density of hexamer of tubes with (−8, 14) symmetry. The SP1 subdomain was modeled using the NMR[6] structure (PDB 2KOD) as a template, and missing residues were modeled using MODELLER[71]. Protonation states of titratable groups were calculated using PDB2PQR[72]. MDFF simulations were performed using the r-RESPA integrator available in NAMD. Long-range electrostatic force calculations employed the PME method (particle-mesh Ewald), utilizing a grid spacing of 2.1 Å and 8th order interpolation, with a 1.2 nm cutoff. The CHARMM36 additive force field was used with a time step of 2 fs, with non-bonded interactions evaluated every 2 fs and electrostatics updated every 4 fs; all hydrogen bonds were constrained with the SHAKE algorithm.

Construction of an immature-like lattice of CA(A92E)–SP1 was performed by docking monomeric HIV-1 CTD (PDB: 2KOD) into the cryo-EM density derived from authentic virions (EMDB: 2706)[15, 39]. All residues present in the NMR-derived structure were used in the initial model. In the same way as in the mature-like case, the SP1 subdomain was modeled using the NMR[6] structure (PDB 2KOD) as a template, and missing residues were modeled using MODELLER[71]. MDFF was applied to the model to refine inter-molecular interactions, and to allow the modeled SP1 subdomain to relax. Additional studies of the SP1 region were performed using the Anton supercomputer.

**Computational docking.** Initial coordinates for BVM were obtained from the open chemistry database (CID: 457928). Hydrogens present in the carboxylic groups of BVM were removed to mimic the protonation states of these groups at physiological conditions. Subsequently, geometry optimization was performed using Gaussian09 using the PM6 semi-empirical method[73], followed by further optimization at a HF/6-31(d) level of theory and final optimization at a B3LYP/6-31(d) level of theory. The optimized structure was then computationally docked into the X-ray derived structure of the immature lattice (PDB: 5I4T) using the HEX protein docking program[74].

**Data availability.** Cryo-EM structural data have been deposited in the EMDB under accession code EMD-8582 and its associated MDFF model in the Protein Data Bank under accession code PDB: 5UP4. Other data that support the findings of this study are available from the corresponding authors upon reasonable request.

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

## Acknowledgements

This work was supported by the National Institutes of Health (NIH Grants P50 GM082251, P41 GM104601, and P50 GM103297 from NIGMS). We acknowledge the support of the National Science Foundation (NSF Grant CHE0959496) for the acquisition of the 850 MHz NMR spectrometer and of the National Institutes of Health (NIH Grants P30GM103519 and P30GM110758) for the support of core instrumentation infrastructure at the University of Delaware; and of the National Institutes of Health (NIH Grant 1S10OD012213-01) for the acquisition of the 750 MHz NMR spectrometer at the University of Pittsburgh. Anton computer time was provided by the Pittsburgh Supercomputing Center (PSC) through Grant R01-GM116961 from the National Institutes of Health. The Anton machine at PSC was generously made available by D.E. Shaw Research. This research is part of the Blue Waters sustained-petascale computing project supported by NSF Awards OCI-0725070 and ACI-1238993, the state of Illinois, and the "Computational Microscope" NSF PRAC Award ACI-1440026. C.M.Q. was supported by National Institutes of Health Postdoctoral Fellowship (F32GM113452). We thank Shannon Modla of the Delaware Biotechnology Institute for assistance with transmission electron microscopy. The cryo-EM data were collected at EICN (Electron Imaging Center for Nanomachines), University of California Los Angeles. We thank DFH Pharma, Inc. for providing the DFH-055 inhibitor.

## Author contributions

T.P., A.M.G., K.S., J.R.P., E.O.F., P.Z. and C.A. designed the study. M.W. and C.M.Q. performed MAS NMR experiments and analyzed the data. R.S. and P.Z. performed structural analysis of CA–SP1 assemblies by cryo-EM. M.W. prepared the CA–SP1 samples of T8I mutant for MAS NMR studies. I.-J.B. performed solution NMR experiments and data analysis. H.Z. and J.R.P. performed chemical shift calculations from the MD trajectories. H.Z. wrote the scripts for dipolar lineshape MAS NMR/MD data analysis. G.H. and C.L.S. performed the MAS NMR experiments and lineshape analysis for the CA–SP1 NL4-3 and A92E assemblies. J.R.P. and K.S. designed and performed the MD simulations. E.O.F., S.A., E.U. provided the CA–SP1 T8I constructs. All authors discussed the results and contributed to the manuscript preparation.

## Additional information

**Competing interests:** The authors declare no competing financial interests.

