## [Peer Review File · Nature Communications]

Reviewers' comments:

Reviewer #1 (Remarks to the Author):

Human immunodeficiency virus (HIV) is the causative agent of acquired immunodeficiency syndrome (AIDS), which largely contributes to the global health burden with an estimated million deaths per year worldwide. Although significant progress has been made towards extended treatment and management of the disease, drug resistance and tolerability are major concerns and more permanent alternative treatment options are of considerable interest. Development of such therapeutics has been hindered by the lack of a thorough structural and mechanistic understanding of HIV pathogenesis, such as the capsid maturation process.

Towards the goal of shining light on HIV capsid maturation at an atomic level, in the present manuscript Wang and colleagues focus on the interface between the capsid protein (CA) and spacer protein (SP1), which has been reported to play a critical role in the HIV maturation pathway. A cryoEM based structural model for the CA-SP1 tubular assemblies is presented, which suggest a lack of a well-defined structure for SP1. Solution NMR and X-ray based docking experiments are performed to characterize the binding of two maturation inhibitors (Bervirimat and DFH-055 methanesulfonate) at the CA-SP1 interface towards a structural basis for their cleavage blocking ability. Magic angle spinning (MAS) Solid State NMR is utilized to characterize an assembly defect rescue mutant CA-SP1(T8I). Interestingly, NMR chemical shift analysis supports the formation of a helical conformation at the CA-SP1 interface. The T8I mutant is additionally shown to promote dynamic and allosteric changes in the CA domain. These findings are complemented by integrated molecular dynamics simulations. Thus, the authors propose that dynamics at the CA-SP1 interface critically regulate the HIV maturation process.

The experiments are performed with rigor and results from complementary biophysical techniques and computations generally support the authors' conclusions. In summary, the data would be of great interest to the protein NMR, HIV biology, and HIV drug discovery communities and are of significant impact for publication in Nature Communications. However, the manuscript could greatly benefit from being revised with a more general audience in mind. Therefore, the manuscript is suggested for publication following a major revision.

Concerns & major points to be addressed by the authors:

1) A major distraction in the narrative of the paper is that the results and discussion were written in the same section and disrupted a nice flow in the manuscript. Furthermore, the manuscript is written with the assumption that the reader is well versed in the HIV Capsid Maturation literature. If the manuscript was slightly tweaked to fix such an assumption it would have greater significance and accessibility to a broader audience.

For example, to the common reader it is unclear why the authors used the CA(A92E) mutation for initial tubular assembly preparations. Please clarify this on pg5 line108 and in the Methods section. Isn't this mutation associated with loss of infectivity in vivo?

2) The authors use DFH-005 methanesulfonate on pg6 line 143 without any reasoning stated. Why characterize the binding of both BVM and DFH-005 methanesulfonate?

3) The figure titles should be more descriptive of the results/significance rather than the experiment performed. It would be helpful if arrows were added to Figure2a showing the direction of chemical shift perturbations to allow the readers to easily follow the titration.

4) Why was the 3.33mM titration point in Figure2a performed at 800MHz when the other points were recorded at 600MHz?

5) A diagram of the residues of CTD-SP1 affected by BVM mapped onto the CTD-SP1 structure (also showing docked BVM) could help show whether the two experiments are in agreement.

6) Rigid body docking of BVM was performed on a previous X-ray structure of CTD-SP1 on pg 6 line 163. Was flexible docking performed? Were the residues affected in NMR titrations used as docking restraints? The authors additionally mention that NMR titrations of assembled CA-SP1 with BVM have so far been unsuccessful (pg 7 line 174). Was docking ever performed on an available CA-SP1 structure? This is a critical point, as any conformational changes that must take place in the bound form would affect the docked structure, and thereby the conclusions of the manuscript.

7) The authors mention poor solubility of the inhibitors precluded KD determination. Have the authors tried to perform NMR titrations under dilute organic solvent conditions to improve molecule solubility (for example, 1% to 5% d6-DMSO)?

8) It would be helpful if the authors could provide some information towards understanding what atoms of BVM are directly affected upon binding to CTD-SP1. Such information would also support the docking models. 1D or 2D cross-saturation transfer experiments by NMR could shed some light on this.

9) A discussion of whether BVM binding to the CTD-SP1 interface is influenced by the proposed

helix-coil equilibrium is lacking. Did the authors test for BVM binding to CA-SP1(T8I)?

10) Is it possible to measure the population of the helix vs coil conformations in each sample?

11) In the introduction (pg 3 lines 66-69) three pathways related to maturation are discussed. In the Conclusion section, the authors should mention whether their new data supports either of these pathways.

Reviewer #2 (Remarks to the Author):

In the manuscript “Stepping on the Brake: Reducing Molecular Motions Modulates HIV Capsid Maturation” by Wang et al. the authors investigate dynamics of CA-SP1 domain of the HIV Gag protein, which is an important intermediate during HIV maturation process. By applying cryoEM, solution NMR, solid state NMR and molecular dynamics simulations, authors try to address the influence of “maturation inhibitors” drugs (bevrimat and DFH-055) and the maturation arresting mutation T8I on the structure and dynamics of CA-SP1. The overall conclusion is that maturation is controlled by modulation of protein dynamics, and that maturation inhibitors perturb this process. This is a generally accepted view in the field, and so with respect to novelty the paper would extend understanding by illuminating the details by which this “modulation” and its “perturbation” occur. Unfortunately, in the opinion of this reviewer the authors use a model system (CA-SP1) that is not appropriate for the questions being posed. In figure 1, the authors present a cryoEM reconstruction showing that CA-SP1 lattice is the mature lattice, thus disproving the implicit assumption made throughout the paper that this is somehow relevant on the conformational changes of SP1. The minor differences in 2-fold or 3-fold packing are really no different from differences observed across different mature lattice structures reported over the years, and so interpretations relating it to the immature lattice are quite a stretch. In figure 2, the authors present bevrimat titration data into soluble CTD-SP1. I do not see the point of this experiment given that it is established that the binding site of bevrimat is the assembled Gag lattice. Overall, there is a serious disconnect between the process being studied (conversion of immature Gag lattice to mature CA lattice), and the model system chosen for analysis.

Responses to Reviewers

Reviewer 1

The experiments are performed with rigor and results from complementary biophysical techniques and computations generally support the authors' conclusions. In summary, the data would be of great interest to the protein NMR, HIV biology, and HIV drug discovery communities and are of significant impact for publication in Nature Communications.

We thank the Reviewer for the interest in our work and for detailed, constructive comments on how to improve our manuscript.

1) A major distraction in the narrative of the paper is that the results and discussion were written in the same section and disrupted a nice flow in the manuscript. Furthermore,, the manuscript is written with the assumption that the reader is well versed in the HIV Capsid Maturation literature. If the manuscript was slightly tweaked to fix such an assumption it would have greater significance and accessibility to a broader audience.

For example, to the common reader it is unclear why the authors used the CA(A92E) mutation for initial tubular assembly preparations. Please clarify this on pg5 line108 and in the Methods section. Isn't this mutation associated with loss of infectivity in vivo?

We very much appreciate this comment and reorganized the manuscript to split the Results and Discussion sections. The CA(A92E) mutation was used since in previous work we showed that the A-E change attenuates the CypA loop dynamics (Lu et al. (2015) PNAS) and the introduction of a charge prevents the tubes from aggregating, resulting in higher resolution cryo-EM data than for the wild type CA. Functionally, this escape mutation renders the capsid non-infectious in the presence of Cyclophilin A, which is normally required for virus infectivity. We added an explanation in the text and the Methods section.

2) The authors use DFH-005 methanesulfonate on pg6 line 143 without any reasoning stated. Why characterize the binding of both BVM and DFH-005 methanesulfonate?

We characterized DFH-005 methanesulfonate in addition to Bevirimat because 1) it is more soluble than Bevirimat, and 2) it is one of the “second-generation” Bevirimat analogs that displays significantly improved antiviral activity relative to the parent compound (e.g., Urano et al. (2016), AAC, 60 (1), p. 190-197). We have added an explanation in the Methods section and in the text.

3) The figure titles should be more descriptive of the results/significance rather than the experiment performed. It would be helpful if arrows were added to Figure2a showing the direction of chemical shift perturbations to allow the readers to easily follow the titration.

We thank the reviewer for this suggestion and have revised the figure captions accordingly.

4) Why was the 3.33mM titration point in Figure2a performed at 800MHz when the other points were recorded at 600MHz?

The titration point at 3.33 was added after the titration curve was initially acquired on the 800 MHz instrument, and at the time the measurement was performed time on the 600 MHz was available. Importantly, the different field strength does not influence the outcome of the results.

5) A diagram of the residues of CTD-SP1 affected by BVM mapped onto the CTD-SP1 structure (also showing docked BVM) could help show whether the two experiments are in agreement.

Thank you for this very helpful suggestion. We have shown the residues experiencing CSPs greater than 0.02 ppm on the CTD-SP1 structure. As is shown in the revised figure, the residues at the CTD-SP1 junction experiencing such large CSPs are those that comprise the binding interface according to the docking simulations. The NMR and docking results are thus in excellent agreement.

6) Rigid body docking of BVM was performed on a previous X-ray structure of CTD-SP1 on pg 6 line 163. Was flexible docking performed? Were the residues affected in NMR titrations used as docking restraints? The authors additionally mention that NMR titrations of assembled CA-SP1 with BVM have so far been unsuccessful (pg 7 line 174). Was docking ever performed on an available CA-SP1 structure? This is a critical point, as any conformational changes that must take place in the bound form would affect the docked structure, and thereby the conclusions of the manuscript.

Rigid body docking was performed as described in the methods section, followed by rigid-body energy refinement. In addition, the resulting models of the complex were subjected to short MD simulations to confirm the stability of the interactions. Thanks to the reviewer, we have extended our rigid-body docking calculations to the recently published CA-SP1 model derived by CryoEM (Schur et al. (2016) Science). The models obtained from the docking procedure to CA-SP1 suggest the same binding modes as the ones obtained for docking of BVM to the structure of CTD-SP1 (Figure S6 shows the in-pore binding mode).

7) The authors mention poor solubility of the inhibitors precluded KD determination. Have the authors tried to perform NMR titrations under dilute organic solvent conditions to improve molecule solubility (for example, 1% to 5% d6-DMSO)?

In the presence of high salt concentration (assembly conditions) the use of organic solvents did not help - the compounds precipitate into the aqueous phase. We are exploring the use of CA-SP1 and Gag constructs where assembly does not require high salt concentrations, and this work will be reported elsewhere.

8) It would be helpful if the authors could provide some information towards understanding what atoms of BVM are directly affected upon binding to CTD-SP1. Such information would also support the docking models. 1D or 2D cross-saturation transfer experiments by NMR could shed some light on this.

Chemical shift perturbations are always ambiguous as to direct or indirect effects and while we see which atoms are affected this could be either due to direct binding or indirect conformational changes upon binding. Due to a limited solubility of BVM (only a fraction of BVM added to the CTD-SP1 solution was soluble), cross-saturation transfer

experiments are not going to work under these circumstances.

9) A discussion of whether BVM binding to the CTD-SP1 interface is influenced by the proposed helix-coil equilibrium is lacking. Did the authors test for BVM binding to CA-SP1(T8I)?

We have added such discussion. BVM will bind to the helical conformer and thereby shift the dynamic equilibrium toward higher content of the helix.

10) Is it possible to measure the population of the helix vs coil conformations in each sample?

This information comes from the analysis of the isotropic chemical shifts recorded in the experiments (which are dynamically averaged) through their calculations in SHIFTX2 on the basis of the MD trajectories. We have performed this analysis for the samples where we have MD trajectories, as illustrated in Fig. 4. The tight agreement between the experimental chemical shifts and those calculated in SHIFTX2 on the basis of the MD trajectories indicates that the populations of helix vs. coil conformations closely resemble those predicted by MD (shown in Fig. 4c). Direct measurement of the populations of the helix vs. coil would require ‘freezing out’ the respective conformers, i.e., cryogenic temperatures. We have performed such experiments; these are reported in Gupta et al. (2016) J. Phys. Chem. B, 120 (2), p. 329-339. Under those conditions we observed the low-populated helical conformer as it has narrow well defined peaks while the predominant coil conformers give rise to wide chemical shift distribution with the lines being broadened beyond detection.

11) In the introduction (pg 3 lines 66-69) three pathways related to maturation are discussed. In the Conclusion section, the authors should mention whether their new data supports either of these pathways.

We thank the reviewer for this comment, we have added a sentence in the Conclusion section. That WT CA-SP1 forms mature lattice suggests that capsid maturation proceeds through either a displacive pathway or a sequential combination of displacive and *de novo* processes.

Reviewer 2

The overall conclusion is that maturation is controlled by modulation of protein dynamics, and that maturation inhibitors perturb this process. This is a generally accepted view in the field, and so with respect to novelty the paper would extend understanding by illuminating the details by which this “modulation” and its “perturbation” occur.

We find this to be an interesting comment because, to our knowledge, our study is the first to unequivocally demonstrate the role of dynamics in the regulation of the structure of the final-step maturation intermediate. The majority of the studies of the immature HIV and maturation intermediates concern static structures of Gag and its products, by cryo-EM, yielding neither atomic resolution picture nor any information on the role of dynamics.

In figure 1, the authors present a cryoEM reconstruction showing that CA-SP1 lattice is the mature lattice, thus disproving the implicit assumption made throughout the paper that this is somehow relevant on the conformational changes of SP1. The minor differences in 2-fold or 3-fold packing are really no different from differences observed across different mature lattice structures reported over the years, and so interpretations relating it to the immature lattice are quite a stretch.

The cryoEM map shows that CA-SP1 adopt a mature lattice in assembly, with a more relaxed CTD trimer interface compared to CA assembly, and disordered or non-helical SP1 segment. We did not suggest that CA-SP1 structure relates to the immature lattice. We realized that the corresponding sentence in the original version of the manuscript may have been confusing and edited it to eliminate any ambiguity.

Unfortunately, in the opinion of this reviewer the authors use a model system (CA-SP1) that is not appropriate for the questions being posed.

Overall, there is a serious disconnect between the process being studied (conversion of immature Gag lattice to mature CA lattice), and the model system chosen for analysis.

Unfortunately, the reviewer took our work out of context. The study focuses on understanding the structure of the final-step maturation intermediate, CA-SP1, which is a *bona fide* maturation intermediate and not a model system (unlike truncated Gag constructs commonly used in the field).

What is currently known in the field is: (1) BVM binds to assembled Gag, indicating that its viral target is an assembled form; (2) BVM binds stably to immature but not mature HIV-1 particles; (3) BVM perturbs HIV-1 maturation by a mechanism that is associated with delayed PR cleavage of CA-SP1 and with stabilization of the immature Gag lattice. What structural effects BVM has on CA-SP1 when it accumulates in the virion is not known. Thus, to assume that binding of BVM to assembled CA-SP1 or understanding the effect of the T8I mutation on the CA-SP1 conformation and dynamics is irrelevant presumes that the structural consequences of BVM binding and T8I mutation are well understood, which is untrue.

Additionally, because of the paucity of structural information regarding the CA-SP1 region of Gag, and the great need for structural information regarding the interaction of BVM with its target, we submit that the structural information in the manuscript represents a critical advance that will be of interest to both drug designers and virologists interested in

maturation inhibitors.

Finally, our work addressed the question of how the T8I mutation in the SP1 peptide mimics the effect of BVM. It appears that BVM stabilizes the helical conformation of the SP1 peptide, and the T8I mutant of the SP1 mimics BVM through stabilization of the helix as well. To our knowledge, our work is the first to *directly* demonstrate this. Moreover, our study is the *first* to provide atomic-resolution information on *both* the structure *and* the dynamics of the SP1 peptide in the assembled last-step maturation intermediate and its T8I mutant, and to reveal the likely binding modes of BVM to the SP1 peptide.

In figure 2, the authors present bevirimat titration data into soluble CTD-SP1. I do not see the point of this experiment given that it is established that the binding site of bevirimat is the assembled Gag lattice.

We respectfully disagree with and are puzzled by this comment which goes contrary to the accepted notions of molecular interactions and specific binding in chemistry, biochemistry, biophysics, and structural biology. Bevirimat *does not* bind to the Gag lattice by recognizing the Gag lattice itself, but by recognizing the SP1 peptide as the specific binding site (see AT Nguen et al. *Retrovirology* (2011), 8:101; **DOI:** 10.1186/1742-4690-8-101). As such, our approach to examine binding of BVM to the CTD-SP1 protein encompassing the entire BVM binding site is warranted; such approach has been validated by decades of research in multiple disciplines and is widely used in drug discovery. Indeed, in the solution NMR binding study we observed that only a selected stretch of residues at the CTD-SP1 junction experience changes upon binding of BVM, and those residues are also the ones that comprise the BVM binding interface according to the docking calculations.

Reviewers' Comments:

Reviewer #1 (Remarks to the Author):

In their revised manuscript, Wang and colleagues successfully respond to previously raised issues with their manuscript, "Stepping on the Break: Reducing Molecular Motions Modulates HIV Caspid Maturation".

First, the authors split the main text into results and discussion sections, which greatly improves the narrative and flow the manuscript. The text now addresses a broader audience. Second, the authors sufficiently address specific concerns brought up regarding technical aspects of their experiments and results.

In summary, the revised manuscript is enthusiastically suggested for publication, pending incorporation of the following minor suggestions:

1. The comment addressing the CA(A92E) mutation on pg 5 should be moved from lines 117-122 to line 100, following the first sentence.
2. The authors use TALOS-N in Table 1 but do not reference it in their manuscript. Please provide the reference: Yang Shen, and Ad Bax, J. Biomol. NMR, 56, 227-241(2013)

Reviewer #2 (Remarks to the Author):

I have carefully read and considered the authors' responses to my comments. I remain of the opinion that the paper extends understanding of the details by which protein dynamics may be modulated during maturation, and how maturation inhibitors perturb this process. With regards to the authors' comment on static structures (to which they contribute in this paper), I would argue that it has been obvious from the dramatically different structures of immature and mature lattices that the dynamic transformations are important – the details of which remain unknown. The authors now clarify that the CA-SP1 structure they study do not relate the immature lattice. However, they do not clearly explain how model system of CA-SP1 tubes (which presumably have related structures to the CA-SP1 cryoEM structure) informs the process of immature-to-mature transition. As the authors say, the CA-SP1 protein is a bona fide intermediate of maturation, but there is no indication in the manuscript on how the CA-SP1 tubes relate to real viruses. Do the tubes represent an an intermediate assembly state during maturation? Are they

simply a useful in vitro system for studying the dynamics of SP1? Esp. given that this journal has a broad audience of non-experts, the authors need to clarify the context in which the study is to be taken, with respect to the dynamics of maturation process that occurs in authentic viruses.

Responses to Reviewers

Reviewer 1

The Reviewer states “In summary, the revised manuscript is enthusiastically suggested for publication, pending incorporation of the following minor suggestions”.

We thank the Reviewer for his/her positive assessment of our manuscript and additional constructive suggestions for improvement. As detailed below, we have revised the manuscript to incorporate the reviewer’s suggestions.

The comment addressing the CA(A92E) mutation on pg 5 should be moved from lines 117-122 to line 100, following the first sentence.

We have moved the comment to line 100, as the reviewer suggested.

The authors use TALOS-N in Table 1 but do not reference it in their manuscript. Please provide the reference: Yang Shen, and Ad Bax, J. Biomol. NMR, 56, 227-241(2013)

We apologize for this inadvertent omission. We have now included the reference in the revised manuscript.

Reviewer 2

We thank the reviewer for carefully reading the revised version of the manuscript and considering our responses to the reviewer’s comments.

With regards to the authors’ comment on static structures (to which they contribute in this paper), I would argue that it has been obvious from the dramatically different structures of immature and mature lattices that the dynamic transformations are important – the details of which remain unknown.

We agree that the dramatically different structures of immature and mature lattices suggest conformational flexibility and dynamic transformations. At the same time, we respectfully disagree with the notion that static structures, particularly at the current level of attainable resolution, are sufficient for understanding biological/biochemical mechanisms in systems where dynamic processes or chemical transformation underlie function. In the case of HIV-1 maturation, static structures of immature and mature lattices represent only the start and end points in the maturation cascade and do not provide any information on the nature of the motions, the timescales or amplitudes of these motions, or on the conformational states and their populations sampled during these dynamic processes. Our study is the first, to our knowledge, to integrate multiple experimental and computational methods for shedding light on these dynamic processes, *quantitatively* and at atomic resolution. Finally, in dynamic systems, cryogenic freezing of the samples may trap only minor (ordered) sub-states along the dynamic trajectory, as we and others have demonstrated (see for example

The authors now clarify that the CA-SP1 structure they study do not relate the immature lattice. However, they do not clearly explain how model system of CA-SP1 tubes (which presumably have related structures to the CA-SP1 cryoEM structure) informs the process of immature-to-mature transition. As the authors say, the CA-SP1 protein is a bona fide intermediate of maturation, but there is no indication in the manuscript on how the CA-SP1 tubes relate to real viruses. Do the tubes represent an intermediate assembly state during maturation? Are they simply a useful in vitro system for studying the dynamics of SP1? Esp. given that this journal has a broad audience of non-experts, the authors need to clarify the context in which the study is to be taken, with respect to the dynamics of maturation process that occurs in authentic viruses.

We thank the reviewer for this important suggestion. In the revised version of the manuscript we have added a paragraph discussing the relationship of the CA-SP1 tubes to the maturation intermediates of the HIV-1 virus and also included a clarification in the abstract. While these tubes are indeed a useful *in vitro* model system to study the SP1 dynamics, we hypothesize that they also represent the features of an intermediate assembly state during maturation. Importantly, dynamics are an inherent feature of SP1, encoded by the primary sequence of the peptide rather than being related to a specific morphology of the assembly. This is very clear from the comparison between the wild type and the T8I mutant of SP1. This finding is based on direct evidence provided by MAS NMR results on the assembled CA-SP1 tubes, from MD simulations on the assembled CTD-SP1 hexamers, as well as additional experimental evidence from solution NMR on the monomeric CTD-SP1 as well as experimental evidence from cryo-EM data of the assembled CA-SP1 tubes. Based on the combined evidences, we submit that the results on SP1 tubes used in our study reflect the behavior of SP1 in authentic viruses. Finally, it is important to note that such model systems capture the essential structural features of the capsids of intact HIV-1 virions, as shown by multiple studies on *in vitro* assembled CA tubes, using different methodologies. Indeed, such types of investigations have provided a wealth of important information which is inaccessible otherwise.